# Long term trends in NHS inpatient bed provision in England, 1960–2020

Patrick Keown[1,2], Ross Alder[3], Georgina Wild[3], Luke Ouma[4], Iain McKinnon [2,5‡], Scott Weich[6‡*]

1 Translational and Clinical Research Institute, Academic Psychiatry, Wolfson Research Centre, Newcastle University, Newcastle upon Tyne, United Kingdom, 2 Cumbria Northumberland Tyne and Wear NHS Foundation Trust, United Kingdom, 3 Newcastle University, School of Psychology, Newcastle upon Tyne, United Kingdom, 4 Biostatistics Research Group, Population Health Sciences Institute, Newcastle University, Newcastle upon Tyne, United Kingdom, 5 Population Health Sciences Institute, Academic Psychiatry, Wolfson Research Centre, Newcastle University, Newcastle upon Tyne, United Kingdom, 6 School of Health and Related Research (SCHARR), Sheffield University, Sheffield, United Kingdom

‡ Professor Weich and Dr McKinnon are joint last authors.
* s.weich@sheffield.ac.uk

## Abstract

### Objectives

To examine the reduction in NHS inpatient hospital beds in England from 1960 until 2020, including five categories: Acute, Geriatric, Maternity, Mental Illness and Learning Disability beds; and to measure regional differences at the end of the study period.

### Design

Retrospective observational study.

### Setting

NHS in England.

### Participants

Inpatient hospital beds.

### Main outcome measures

NHS inpatient bed provision per 100,000 population. Rate of reduction calculated as percentage change each year. NHS bed provision in 7 regions of England compared for the year 2019/20.

### Results

NHS inpatient bed provision declined for sixty consecutive years. The overall reduction was 78.0% between 1960 and 2020. Greatest reduction was in Learning

**Data availability statement:** All relevant data are within the manuscript and its supporting information files.

**Funding:** The author(s) received no specific funding for this work.

Disability beds (98.7%), followed by Mental Illness (90.6%), Geriatric (75.0%), Maternity (67.4%) and the least reduction in Acute beds (63.0%). There were two periods of accelerating rates of bed reduction, the first in the 1980s, and the second in the 2000s. At the end of the study period there was significant regional variation in bed numbers.

## Conclusions

Bed reductions were a constant feature, with important differences between bed categories and across time. This needs to be addressed when planning for future pandemics and winter bed pressures. By the end of the study the NHS was no longer providing the same level of inpatient care in different regions of England, particularly for Learning Disability.

## Introduction

Hospital inpatient bed numbers have reduced in many health care systems across the world particularly in established market economies [1] where there has been a drive for efficient use of inpatient beds, partly by reducing length of stay. The latest available data from the Organisation for Economic Cooperation and Development (OECD) indicate that the United Kingdom has the lowest number of inpatient hospital beds per capita of any of the G7 nation (2.3 per 1,000 population), less than half the number of France (5.7) or Germany (7.8), and less than a fifth of the number in Japan (12.6) [2].

The National Health Service (NHS) provides the majority of inpatient beds in the United Kingdom (UK) with a relatively small but increasing proportion being provided in the private health care sector. The NHS provides a comprehensive range of inpatient hospital services including Medical, Surgical, Geriatric (care of the elderly), Obstetric, Paediatric, Mental Illness and Learning Disability (intellectual disability). It is estimated that there were between 480,000 beds in England and Wales and 550,000 beds across the UK in at the inception of the NHS [3,4] and the number of Mental Illness and Learning Disability beds peaked at 152,000 in 1954 [5]. Since then, bed numbers have reduced across the NHS and these reductions have not occurred uniformly [6]. However, previous research on bed numbers in the NHS has tended to focus on time periods lasting ten, twenty, or thirty years, and on particular types of inpatient hospital beds including psychiatric beds [7–9].

One measure of the adequacy of inpatient beds that has been used to compare health services in different countries is the ratio of deaths to inpatient beds [10]. This ratio helps account for the "nearness to death effect" on hospital bed use which significantly impacts demand and therefore provides more accurate international comparison than simply looking at beds per capita, as it acknowledges that a large proportion of hospital bed usage occurs in the last year of life. There were particular concerns about how the NHS would cope with the surge in demand for inpatient care during the waves of COVID-19. In the years leading up to the COVID-19 pandemic it

was clear that the NHS bed capacity was insufficient as each winter brought a bed crisis necessitating the redeployment of staff to emergency care and the cancelling of operations and outpatient appointments [11]. When the COVID-19 pandemic arrived, each nation state in the UK sent out a message to the public to "protect the NHS" [12]. The aim was to slow the spread of COVID-19 so that hospitals and clinics across the country were not faced with more patients than they were able to treat, and so were able to continue delivering care.

The aim of this study is to quantify the change in the total number and different categories of NHS inpatient hospital beds between 1960 and 2020 in England, and to test the null hypothesis that the annual reduction in beds occurred uniformly across the 60-year period and affected all sectors/bed types equally. In addition to compare these changes with the level of demand for inpatient care including the number of births and deaths in England. Variation between regions in bed provision will be compared in 2019/20.

## Materials and methods

The main source of data on inpatient bed numbers was the Bed Availability and Occupancy statistical work area on the NHS England website [13] with data available from 2008/9–20119/20. The numbers of inpatient beds in England between 1960 and 2008 were obtained from NHS England following a data request. The average daily number of NHS beds in each calendar year was available between 1960 and 1986. From 1987/8 central health data returns (known as Korner Returns) were introduced. This included the KHO3 metric, which was an annual return of the average daily number of available beds by ward classification for each financial year from 1987/88 onwards. Day Only beds were also included from 1987/8. Day Only beds are located on wards that are open during the day but not overnight and typically are used for day surgery cases or other procedures that do not require overnight stay.

In 2010/11 this changed to a quarterly collection of the average number of available NHS beds by consultant main specialty. Intermediate care beds and residential care beds were no longer included from 2010/11 onwards. These included intermediate care beds in community hospitals and care homes that did not directly involve a consultant. This change affected the data for the year 2010/11 with minimal impact on subsequent years. Most of the residential care beds were within the Mental Illness and Learning Disability category and most intermediate care beds were in the General & Acute category [14].

Bed data were collected in four categories provided by the NHS for the whole of the study period: General & Acute, Maternity, Mental Illness, and Learning Disability. "All NHS inpatient beds" was the sum of these categories. The General & Acute category included Medical, Geriatric, Surgical and Paediatric beds. Data were available separately for Geriatric beds and Acute (Medical, Surgical and Paediatric) beds from 1960 until 2009/10. From 2010/11 these were combined in the General & Acute category. However, the number of occupied beds (compared with available beds above) was still recorded for Geriatric Medicine. Assuming an occupancy of 91% for Geriatric beds (based on 10 years of data prior to 2010), the number of occupied beds was multiplied by 1.1 to estimate the number of available Geriatric beds. This figure was subtracted from the number of General & Acute beds to estimate the number of Acute beds.

Data were also obtained from the Hospital Admitted Patient Care Activity module of Hospital Episode Statistics on Finished Consultant Episodes (FCEs), Admissions and Occupied Bed Days [15]. This module includes all episodes of care for patients admitted to hospital including days cases and critical care. It does not include out-patient appointments or accident & emergency attendance. FCEs are the number of episodes of inpatient care in a year. Occupied bed days are the total number of days of inpatient care under a consultant in a year. Length of stay is the number of days of inpatient care between admission and discharge. This data was available from 1998/9. Data was analysed for whole NHS in England and for the 15 Consultant Specialities that accounted for the greatest number of Occupied Bed Days in 1998/9. The organisations providing NHS funded inpatient hospital care was also available. In 1998/9 all this care was recorded as being provided by 348 NHS organisations. By 2019/20 some of this inpatient care was being provided by the Independent Sector (83 organisations) as well as by 237 NHS organisations. Many organisations in the NHS and Independent

Sector operated more than one hospital but the precise number of hospitals is not available. Only organisation providing inpatient care were included. There was no data available for some specialities regarding the involvement of the Independent Sector including psychiatric specialities. Therefore, other sources of information were used including statutory reports from NHS England: Mental Health Act Statistics, Annual Reports [16]; and Learning Disability Statistics [17]; and a report from the Care Quality Commission on Inpatient Psychiatric Rehabilitation services [18].

Population data for England from 1971–2020 were obtained from the Office for National Statistics [19]. The combined population for England and Wales was available for 1960–1970. We estimated the population of England alone for the years 1960–1970 by estimating that 94.35% of the combined population was in England. This was based on data from 1971–1980 which recorded populations for both England and Wales separately. Bed data and the number of deaths were available for seven regions in England for the year 2019/20. These regions were North West, North East and Yorkshire, Midlands, East, South West, South East and London. The population estimates for these Regions were obtained for 2019.

The Gross Domestic Product and annual changes were available for the UK from the Office of National Statistics and were used as a measure of economic activity [20]. The annual number of deaths and of births in England were obtained from the Office of National Statistics [21].

## Statistical analysis

NHS inpatient bed provision was calculated as the number of inpatient beds per 100,000 population. Rates for Geriatric beds were calculated per 100,000 total population and per 100,000 of the population aged 65 years and older. Annual changes in bed numbers and bed provision, and annual percentage reduction in bed numbers were calculated. The annual percentage change (APC) in bed provision was the variable used to measure rate of change. This was calculated as the percentage change from one year to the next using population adjusted rates for each year and each bed category, along with the average (median and mean), standard deviation (SD) and range.

To fully ascertain differences in bed capacity over time, including evidence of nonlinearity in changes in bed capacity over time, we explore a non-linear association between year (continuous variable) and annual rate of reduction using a restricted cubic spline (RCS).

RCS model non-linear relationships by fitting piecewise cubic polynomials between predefined knots. They divide the range of the data into sections using points called knots and fit smooth curves (cubic polynomials) between these sections [22]. To prevent overfitting and improve interpretability, the model is 'restricted' (i.e., constrained) to be linear at the extremes (beyond the first and last knots). The number of knots for the RCS were obtained by fitting several spline models and comparing their AIC/BIC (Akaike Information Criterion/ Bayesian Information Criterion), to determine which number of knots gave the best model fit. We also explored the suitability of other non-linear models including fractional polynomials, and non-parametric (LOESS), but RCS outperformed them. LOESS (Locally Estimated Scatterplot Smoothing) is a non-parametric regression technique that fits a smooth curve to a set of data points by using local weighted regression. Instead of assuming a single global function, LOESS fits many local regressions using weighted least squares, giving more influence to points near the target value, resulting in a smooth curve that closely follows the data trend without imposing a specific functional form". These non-linear models help explore associations between the variables and outcome, but we do not use them to infer causality.

The annual reduction in NHS bed numbers in England were compared with the annual change in Gross Domestic Product in the UK to explore the association with economic activity. The ratio of births to NHS Maternity beds was calculated along with the ratio of deaths to the number of NHS General & Acute beds (1,000s) as a measure of the level of demand on beds.

We further explored for any differences in bed type numbers across seven regions in England. Here, we fitted a Poisson model with region as a predictor variable, population size as an offset to adjust for regional differences in population. All statistical analysis was performed in R software (version 4.3.0). The mortality rate in each of the seven regions was plotted against the number of NHS beds per 1,000 deaths in each region.

## Results and discussion

### Changes in England 1960–2019/20

There was a decline in bed numbers across all bed categories and in the total number of NHS inpatient beds between 1960 and 2019/20. This is illustrated in Fig 1. There were no years during which the total provision of NHS inpatient beds increased, although for some bed types such as maternity there was an increase in a limited number of years (1960–1967) or no change for a prolonged periods (geriatric beds 1960–1984). In other categories there were continuous reductions.

The size of the reduction in NHS inpatient bed was 71.3% (319,241 beds). Numbers went from 447,457 in 1960–128,216 in 2019/20 in England. As the population of England increased by 30.3% from 43.2 million to 56.3 million during this time, the provision of inpatient beds per 100,000 reduced by 78.0%, from 1,036–228 beds per 100,000. This reduction occurred in all bed categories but with substantial differences between categories. Learning Disability had the highest percentage reduction in bed capacity (98.7%), followed by Mental Illness (90.6%), Geriatric (75.0%), Maternity (67.4%) and the least reduction was in Acute beds (63.0%). If the rate of Geriatric beds is calculated per 100,000 of the population aged 65 years and older then the reduction was 84.3%. See Supplementary S2 Table in S1 File.

Fig 1 shows that at the start of the study the NHS had a balanced portfolio of inpatients beds in the five different categories. By the end of the study this had changed to a situation where inpatient bed services were dominated by the Acute category, with the remaining five categories (Mental Illness, Geriatric, Maternity and Learning Disability) making up just over a third of inpatient beds.

At the start of the study Acute beds and Mental Illness beds accounted for 38.5% and 32.2% of NHS bed provision respectively. The largest reductions in number of beds were also in these categories, however the rate of reduction was

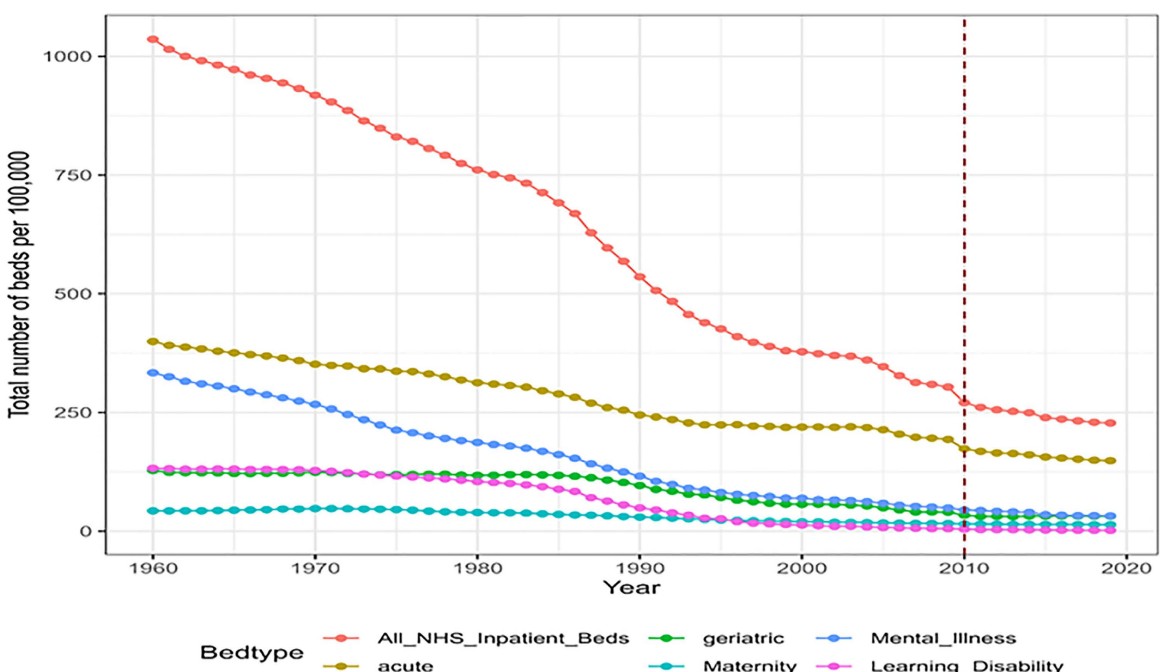

**Fig 1. Changes in NHS bed numbers per 100,000 between 1960 and 2019 in England. The red dashed line represents the implementation of a new recording system in 2010/2011.**

quite different. See Fig 1. As a consequence, by the end of the study Acute beds accounted for 65.2% of NHS bed provision and Mental Illness beds accounted for only 14.2%.

Learning Disability beds were the third largest category in 1960, accounting for 12.8% of NHS bed provision. In Fig 1 the line for Learning Disability beds crosses below Geriatric beds in the 1970s and crosses below Maternity beds in the 1990s. Thus by the end of the study period Learning Disability beds had become the smallest category accounting for just 0.7% of NHS bed provision.

Maternity beds were by far the smallest bed category at the start of the study, accounting for 4.1% of total NHS bed provision. Maternity beds had the smallest number of beds closed during this period. By the end of the study Maternity beds accounted for 6.1% of NHS bed provision and no longer were the smallest category.

Geriatric beds accounted for 12.3% of NHS bed provision in 1960. There was no overall reduction in Geriatric beds in the first twenty-five years of the study. As a result, during the middle of the study period Geriatric beds accounted for nearly 25% of NHS bed provision. Geriatric bed reductions started in the 1980s and then continued. By the end of the study Geriatric beds accounted for 13.9% of NHS bed provision.

### Day only beds

Data on Day Only beds were available from 1987/88 when it is recorded there were exactly 2,000. This increased steadily to 12,762 by 2019/20, a 583.1% increase. The rate of Day Only beds per 100,000 population increased from 4.2 to 22.7. During this 23-year period nearly 16 inpatient beds were closed for every Day Only bed created. If it is assumed that there were no Day Only beds in 1960, then over the whole study period 25 inpatient beds were closed for every Day Only bed created.

### Annual rates of reduction in NHS bed provision

NHS bed provision reduced by 2.4% each year on average (median 2.2%, range 0.4% to 6.0% reduction). In Fig 2, we report the annual percentage reduction (APC) in the provision of All NHS beds. Here we notice a pattern of lower rates of bed closures in the in 1960s and 1970s followed by higher rates starting in the late 1980s. This pattern was evident in most bed categories except Maternity beds which had an increase in bed provision in several years in the 1960s and then a modest rate of reduction starting in the 1970s and persisting until the end of the study.

We also notice the spike in APC in All NHS beds and Acute, Geriatric and to lesser degree Maternity categories for the year 2010/11, which is attributable to the change in the recording process for NHS inpatient beds. This was due to intermediate care beds and residential care beds that the NHS was still operating no longer being included in the inpatient bed numbers from that year onwards. Although the percentage reduction in bed numbers across the years was highly variable across the six bed types (Fig 2), we notice that in the last 5–6 years (2014/5–2019/20), there was another period of low rates of bed closures.

Fig 3 shows a non-linear association between year and annual percentage reduction in All NHS bed provision, modelled using a restricted cubic spline (RCS). We explored various number of knots, between 3 and 9, using AIC and BIC, 7 was the optimal number of knots chosen for the analysis. Model outputs are available in Supplementary S2 Table in S2 File. The visualisation demonstrates evidence of two specific periods of accelerating rates of bed closures. The first period of accelerating bed closures was 1980–1990. The second period was 2000–2010.

We may draw a similar inference from Fig 1, noting that in the period with the highest accelerated bed closures (1980–1990), almost 250 units of bed provision were lost in this 10-year window. This was close to 30% of the reduction in bed provision that occurred over the 60-year study period, with nearly 20% occurring in the second half of that decade.

As shown in Fig 3, it is clear that the annual rate of reduction in beds across the years differed significantly between different time periods. For instance, annual reduction rates in the 1960s and 1970s (mean & median = 1.5%) were less than half the rates in the 1980s and 1990s (median = 3.1 & mean = 3.4). Rates in the last two decades of the study were in between the two (2000s & 2010s, median = 1.6 & mean = 2.6). See Fig 3.

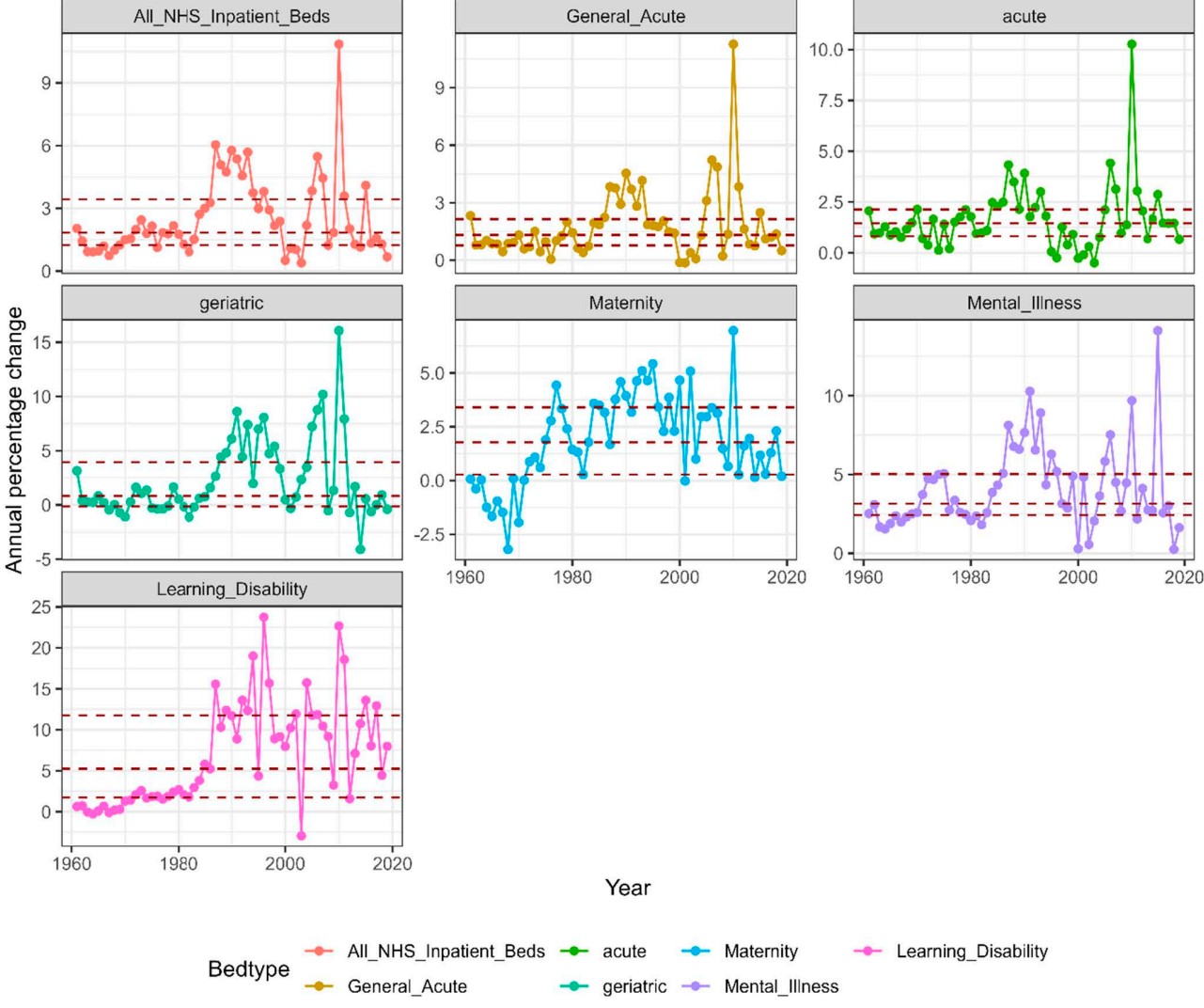

**Fig 2. Annual percentage reduction in all NHS inpatient bed provision per 100,000 population between 1960 and 2019/20 in England.** The red horizontal lines represent the 25th, 50th (median) and 75th percentiles.

## Annual change in GDP and NHS beds

GDP in the UK increased from £572 million in 1960 to £2.234 billion in 2019/20. The average annual change was +2.4% and ranged from −4.6% to +6.5%. There were seven years of negative growth. The number of NHS beds fell from 447,457–128,216. The average annual change was a 2.5% reduction in beds, ranging from a 0.4% reduction to a 10.9% reduction. There was no association between the annual change in GDP and the annual change in NHS beds (Pearson correlation r = 0.00, N = 59, p = 0.98). See Fig 4. The average annual reduction in beds was 2.2% in the seven years of negative growth in the economy compared with an average reduction of 2.6% in years when the economy grew.

## Changes in births and deaths relative to NHS beds

The number of births fluctuated during the study period peaking at 828,470 in 1964 and dropping to 536,953 in 1977 before rising again, with an average of 655,334 per year. The ratio of births to NHS Maternity beds increased from 40.0 in

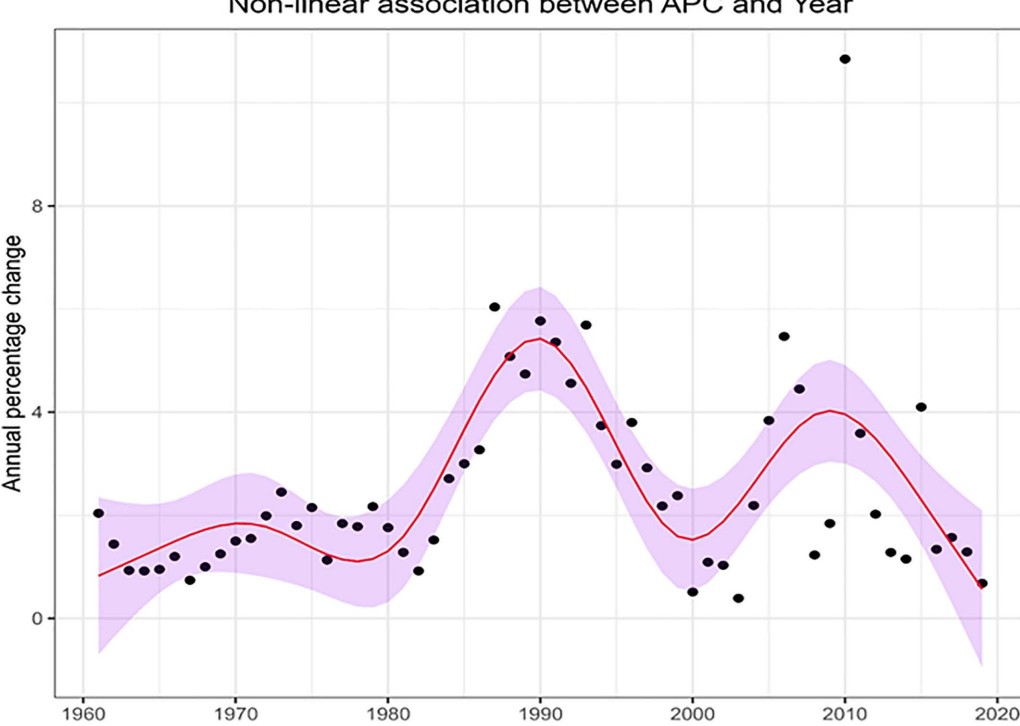

**Fig 3. Non-linear association between annual percentage reduction in total NHS bed numbers and year. These are results of a restricted cubic spline model with seven knots**. The shaded region represents the 95% confidence bands of the prediction, the red line the model fit.

1960 to 79.6 at the end of the study period (ratio was lowest in 1976 at 26.6 and highest in 2012 at 88.4) See Supplementary S2 Fig S1 File.

The number of deaths each year was relatively stable with an average of 516,723 and ranging from 452, 862–560,317. The ratio of deaths to General & Acute beds steadily increased from 2.2 in 1960 to 4.9 in 2019/20. The rate of increase accelerated: it was 28 years before the ratio increased from 2.2 in 1960 to rising above 3.0 in 1988/9; 22 further years before this ratio went above 4.0 in 2010/11; but only 8 further years before the ratio reached 5.0 in 2018/9. See Supplementary S2 Fig S1 File.

### Changes in admissions, length of stay and occupied bed days

The number of episodes of inpatient care (FCEs) rose by 75% between 1998/9 and 2019/20: increasing from 11 million episodes to 21 million. In 1998/9, the average age of patients was 45 years and 57% were recorded as female gender. By 2019/20 average age had increased to 54 years, and 55% were recorded as female. The largest increase in episodes of care was in the elderly with a 147% increase for those aged 75 + years (2.2 to 5.5 million FCE) compared with 20% increase in those aged 0–14 years (1.7 to 2.0 million FCEs).

At the same time Occupied Bed Days reduced by 11% from just over 50 million days to just under 45 million days. This was achieved by reducing the average length of inpatient admissions by half: median length of stay from 2 days to 1 day; mean length of stay from 8.4 to 4.5 days.

The increase in inpatient activity was marked (>50%) in 4 specialities: Cardiology; Geriatric Medicine; General Medicine; and Oncology. There were moderate increases in 5 specialities: Orthopaedic Surgery: General Surgery; Paediatrics;

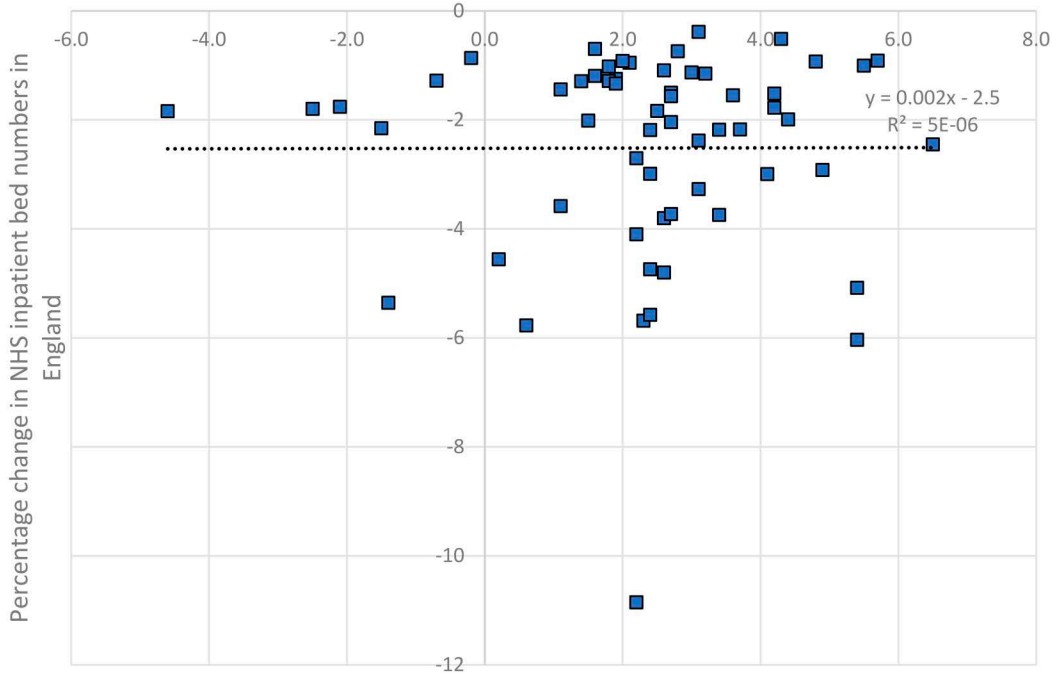

Year on Year percentage change in UK economy (GDP)

$y = 0.002x - 2.5$
$R^2 = 5E{-}06$

**Fig 4. Annual percentage reduction in NHS bed numbers in England and the year-on-year change in the UK economy (GDP).** Dotted line is linear trend, with linear regression equation also shown with $R^2$ value.

**Table 1. Comparison of inpatient activity in England for the NHS and 15 hospital specialities that accounted for the largest number of Occupied Bed Days. Data from 1998/9 are compared with 2019/20.**

| Speciality | | Finished Consultant Episodes | | | Median length of stay (days) | | | Number of Occupied Bed Days | | |
|---|---|---|---|---|---|---|---|---|---|---|
| Code | Name | 1998/ 1999 | 2019/ 2020 | % change | 1998/1999 | 2019/2020 | % change | 1998/ 1999 | 2019/ 2020 | % change |
| 300 | General Medicine | 2,040.0 | 3,450.6 | +69% | 4 | 1 | −75% | 9,544,976 | 7,869,542 | −18% |
| 430 | Geriatric Medicine | 555.6 | 1,044.8 | +88% | 11 | 6 | −45% | 8,336,356 | 6,464,556 | −22% |
| 710 | Mental Illness | 160.6 | 77.0 | −52% | 17 | 20 | +18% | 5,310,720 | 2,678,968 | −50% |
| 100 | General Surgery | 1,430.3 | 2051.6 | +43% | 3 | 1 | −67% | 4,587,115 | 3,606,893 | −21% |
| 110 | Trauma & Orthopaedic | 799.1 | 1,163.6 | +46% | 3 | 2 | −33% | 4,182,561 | 3,402,222 | −19% |
| 420 | Paediatrics | 1,097.4 | 1,500.6 | +37% | 1 | 1 | 0% | 2,739,423 | 2,621,876 | −4% |
| 715 | Old Age Psychiatry | 51.8 | 13.3 | −74% | 22 | 57 | +159% | 2,079,125 | 795,327 | −62% |
| 502 | Gynae -cology | 1,093.3 | 774.0 | −29% | 1 | 1 | 0% | 1,685,893 | 852,377 | −49% |
| 501 | Obstetrics | 780.1 | 749.3 | −4% | 1 | 1 | 0% | 1,563,392 | 1,120,520 | −28% |
| 101 | Urology | 567.2 | 712.0 | +26% | 3 | 1 | −67% | 1,005,423 | 709,653 | −29% |
| 320 | Cardiology | 219.0 | 707.8 | 223% | 3 | 3 | 0% | 716,229 | 1,712,847 | +139% |
| 700 | Mental Handicap | 38.7 | 5.5 | −86% | 3 | 3 | 0% | 583,877 | 140,787 | −76% |
| 170 | Cardio-thoracic | 66.8 | 77.2 | +16% | 7 | 6 | −14% | 502,482 | 544,533 | +8% |
| 120 | Ear, Nose & Throat | 382.4 | 346.8 | −9% | 1 | 1 | 0% | 484,170 | 307,950 | −36% |
| 800 | Clinical Oncology | 276.0 | 427.4 | +55% | 3 | 2 | −33% | 444,976 | 280,030 | −37% |
| | **Total (all specialities)** | **11,983.9** | **20,912.3** | **+75%** | **2** | **1** | **−50%** | **50,294,678** | **44,820,576** | **−11%** |
| | % change | Marked reduction > 50% reduction | | Moderate reduction 50% to 0% reduction | | Moderate increase 0 to +49.9% increase | | Marked increase > 50% increase | | |

Urology; and Cardiothoracic surgery. In contrast there was a moderate reduction in 3 specialities: Gynaecology, ENT surgery; and Obstetrics. And a marked reduction in 3 specialities: Mental Handicap, Mental Illness and Old Age Psychiatry. See Table 1. Median length of stay reduced or remained the same in 13 specialities but increased in 2 specialities: Old Age Psychiatry and Mental Illness. The net result of these changes was that in the majority of specialities (13 out of 15) the number of Occupied Bed Days reduced, with the greatest reduction occurring in the 3 psychiatric specialities of Mental Handicap, Old Age Psychiatry and Mental Illness. In contrast there was a modest increase in Cardiothoracic and a marked increase in Cardiology Occupied Bed Days.

### Role of the independent sector

The specialities that experienced the greatest reductions in NHS inpatient episodes of care between 1998 and 2019/20 were the same specialities that showed the highest involvement of the Independent Sector by 2019/20. There were three broad groupings in this regard. Psychiatric services had the greatest reduction in episodes of inpatient care (>90%) and had the highest provision by the Independent sector (23.7% to 52.5%). Surgical specialities and Obstetrics & Gynaecology had moderate changes to inpatient episodes of care (ranging between 32% reduction and 37% increase) and modest provision by the Independent Sector (0.04%−16.3%). Medical specialities had a marked increase in NHS inpatient care episodes (increase between 69% to 223%) and the lowest provision by the Independent sector (0.07%−0.14%). See Table 2 and Fig 5.

### Variation between regions of England

There was a modest degree of variation in most NHS inpatient bed provision between the regions in England in 2019/20. There was a less than twofold variation between regions in nearly all bed categories. The exception was Learning

**Table 2. Estimates of the percentage of NHS funded inpatient care being provided in the Independent Sector.**

| Data source | Finished Consultant Episodes (1,000s) in the financial year 2019/2020 provided by NHS hospitals and Independent Sector hospitals | | | | | | | | | Inpatients detained Mental Health Act on 31st March 2020. | Psychiatric rehab inpatient beds 2019 | Inpatients 1° diagnosis Learning Disability Autism March 2020 |
|---|---|---|---|---|---|---|---|---|---|---|---|---|
| Speciality | General Medicine and related specialities | | | General Surgery, related specialities and Gynaecology | | | | | | Psychiatry | | |
| Medical speciality | General Medicine | Geriatric Medicine | Cardiology | General Surgery | Urology | Ear, Nose & Throat surgery | Trauma & ortho | Cardiothoracic | Gynaecology | Adult, Old Age, Forensic, Child & Adolescent, & Learning Disability | Psychiatric rehabilitation (part of Adult Psychiatry) | Learning Disability |
| NHS | 3437.6 | 1032.3 | 706.4 | 1963.8 | 684.5 | 337.0 | 984.2 | 77.2 | 744.7 | 11,055 | 2,077 | 1,050 |
| Independent sector | 3.1 | 0.7 | 0.9 | 79.3 | 25.4 | 9.7 | 173.0 | <0.1 | 27.3 | 3,450 | 2,295 | 1,005 |
| Total | 3440.7 | 1033.0 | 707.8 | 2043.2 | 709.8 | 346.7 | 1157.2 | 77.2 | 772.9 | 14,555 | 4,372 | 2,095 |
| % Independent sector | 0.09% | 0.07% | 0.14% | 3.87% | 3.52% | 2.88% | 16.31% | 0.04% | 3.5% | 23.70% | 52.48% | 47.97% |
| Change in FCEs in NHS hospitals from 1998/9 | +69% | +86% | +223% | +37% | +21% | −12% | +23% | +16% | −32% | −60% | | −86% |

No data was available for the number of FCEs provided by Independent Services in 2019/20 for Paediatrics, Clinical Oncology, Obstetrics, Mental Illness, Old Age Psychiatry, Mental Handicap. Alternative data was available for Psychiatric Services and these are presented the column on the right hand side of the table. No alternative data was available for Paediatric, Clinical Oncology and Obstetrics so these have been excluded.

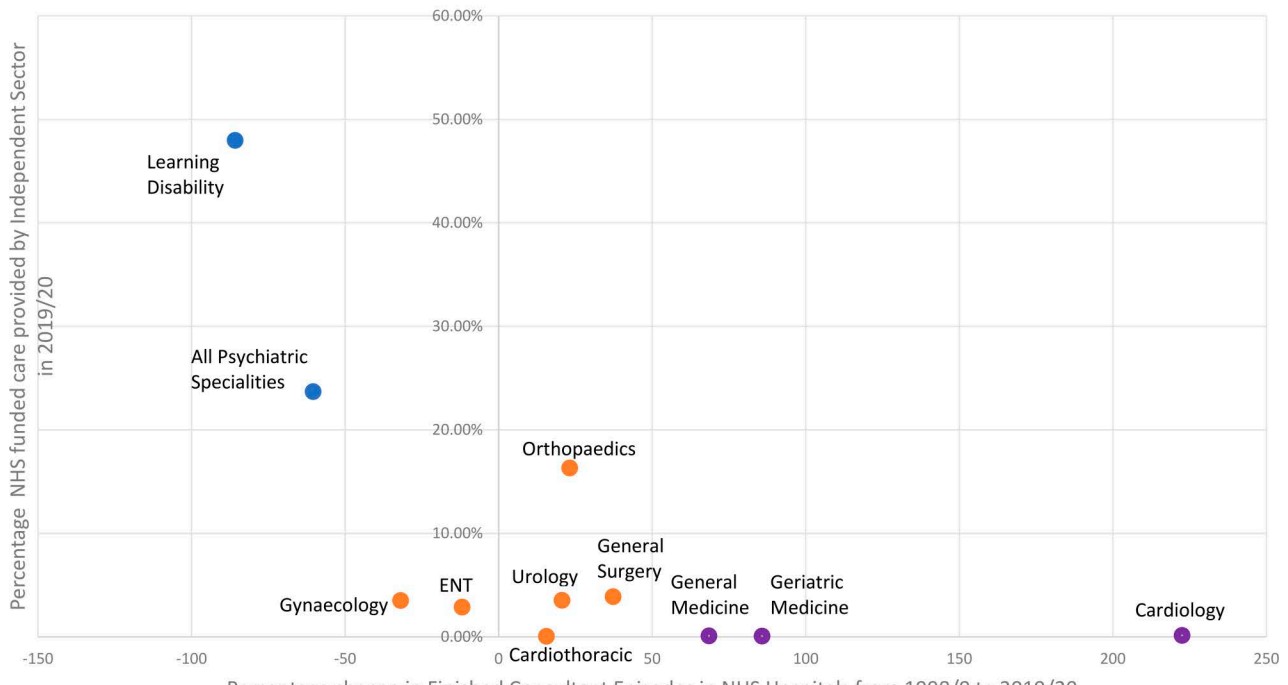

**Fig 5. Change in the number of Finished Consultant Episodes in NHS hospitals between 1998/9 and 2019/20 and the percentage of NHS funded inpatient care provided by the Independent Sector in 2019/20.** Data is for 11 medical specialities in England.

**Table 3. The provision of NHS inpatient beds per 100,000 population in England in 2019/20. Seven NHS regions are shown separately (North East, North West, Midland, East, South West, South East and London). Total NHS bed numbers are shown along with five categories: Acute; Geriatric; Maternity; Mental Illness; and Learning Disability.**

| NHS commissioning Region | All beds | Acute | Geriatric | Maternity | Mental illness | Learning Disability |
|---|---|---|---|---|---|---|
| North East | 249.83 | 156.9 | 39.92 | 13.95 | 35.39 | 3.69 |
| North West | 260.25 | 183.5 | 28.66 | 14.28 | 31.35 | 2.47 |
| Midlands | 227.82 | 155.69 | 25.18 | 12.86 | 31.94 | 2.15 |
| East | 202.26 | 132.16 | 30.39 | 14.31 | 23.93 | 1.48 |
| South West | 215.28 | 141.8 | 35.44 | 10.7 | 27.05 | 0.29 |
| South East | 190.47 | 120.04 | 32.78 | 10.68 | 26.35 | 0.63 |
| London | 237.91 | 144.38 | 30.59 | 17.55 | 44.79 | 0.6 |
| Variation between lowest and highest provision of each bed category | 36% SE vs. NW | 53% SE vs. NW | 59% Midlands vs.NE | 64% SE vs. London | 87% East vs London | 1,172% SW vs. NE |

Disability beds where there was a greater than twelvefold variation between regions (3.7 per 100,000 in North East compared with 0.3 in South West). NHS inpatient bed provision in London showed a distinct pattern with the highest rates of both Mental Illness bed provision (87% higher in London than in the East of England) and Maternity bed provision (64% higher in London than in the South East). See Table 3.

On adjusting for population differences across the regions, although total inpatient bed numbers were significantly different between London and nearly all other regions (East, Midlands, North West, South East and North West), the

**Table 4. Comparison of NHS inpatient bed numbers across regions in England in 2019/20. Results are estimates from a Poisson model adjusting for differences in population across the various regions. Standard errors of the coefficients are also shown. Comparisons across regions are made with London as the reference region. Total bed numbers are compared as well as five categories: Acute; Geriatric, Maternity, Mental Illness; and Learning Disability. (\*\*<0.05 significance level, \*\*<0.01 significance level, and \*\*\*<significance 0.001).**

| NHS commissioning Region | All beds Est. | p value | Acute Est. | p value | Geriatric Est. | p value | Maternity Est. | p value | Mental illness Est. | p value | Learning Disability Est. | p value |
|---|---|---|---|---|---|---|---|---|---|---|---|---|
| North East | 0.14 (0.09) | 0.118 | 0.18 (0.12) | 0.121 | 0.35 (0.24) | 0.147 | 1.48 (1.12) | 0.186 | −0.16 (0.36) | 0.656 | −0.16 (0.23) | 0.481 |
| North West | 0.30 (0.09) | **0.001**\*\* | 0.45 (0.11) | **<0.001**\*\*\* | 0.13 (0.26) | 0.605 | 0.89 (1.22) | 0.466 | −0.05 (0.36) | 0.886 | −0.17 (0.23) | 0.461 |
| Midlands | −0.23 (0.09) | **0.014**\* | −0.11 (0.12) | 0.366 | −0.40 (0.27) | 0.137 | 0.51 (1.22) | 0.678 | −0.51 (0.36) | 0.161 | −0.53 (0.23) | **0.023**\* |
| East | 0.20 (0.01) | **0.039**\* | 0.28 (0.12) | **0.023**\* | 0.33 (0.26) | 0.199 | 0.36 (1.41) | 0.798 | 0.11 (0.36) | 0.760 | −0.27 (0.25) | 0.291 |
| South West | 0.36 (0.09) | **<0.001**\*\*\* | 0.45 (0.12) | **<0.001**\*\*\* | 0.59 (0.25) | **0.018**\* | −21.83 (1.45) | 1.000 | −0.03 (0.38) | 0.940 | −0.05 (0.24) | 0.848 |
| South East | −0.25 (0.10) | **0.011** | −0.20 (0.12) | 0.096 | 0.04 (0.25) | 0.876 | −0.02 (1.41) | 0.987 | −0.52 (0.38) | 0.178 | −0.57 (0.25) | **0.020**\* |
| London | reference | | reference | | reference | | reference | | reference | | reference | |

**Significance levels:** \* *p* < 0.05; \*\* *p* < 0.01; \*\*\* *p* < 0.001

differences in NHS inpatient bed numbers is only visible across particular bed types. See Table 4. On examining the bed type differences across regions, we observed no significant differences in Maternity and Mental Illness bed numbers in all regions compared to London. Similar observations were made for Geriatric beds except for South West. For Acute inpatient beds, we noticed statistically significant differences between London and East. In general, it appears differences in total inpatient bed numbers between London and other regions were largely driven by the differences in Acute and Learning Disability bed numbers. It is also interesting to note that bed numbers in the North East and London were relatively similar, except for differences in general acute bed numbers.

The average mortality rate was 8,845 deaths per 100,000 population in the seven regions in 2019/20. This rate was much lower in London (5,427, N = 1) compared with the rest of the country (9,334, N = 6). The average number of NHS inpatient beds per 1,000 deaths was 268 in the seven regions. This ratio was much higher in London (440, N = 1) than in the rest of the country (239, N = 6). There was a strong negative association between these two measures (r = −0.9) with linear regression indicating that the mortality rate was lower by 1 unit for every 18 NHS inpatient beds per 1,000 deaths. See Fig 6.

## Discussion

There were sixty consecutive years of reduction in NHS inpatient capacity in England prior to the onset of the COVID 19 pandemic. It is striking that there was no single year when NHS inpatient bed provision increased. These reductions disproportionately affected Learning Disability, Mental Illness, and Geriatric beds. Whilst at the start of the study the NHS had a balanced portfolio of the five inpatient bed categories, by the end of the study the Acute category heavily dominated NHS inpatient services, accounting for more than two thirds of inpatient beds.

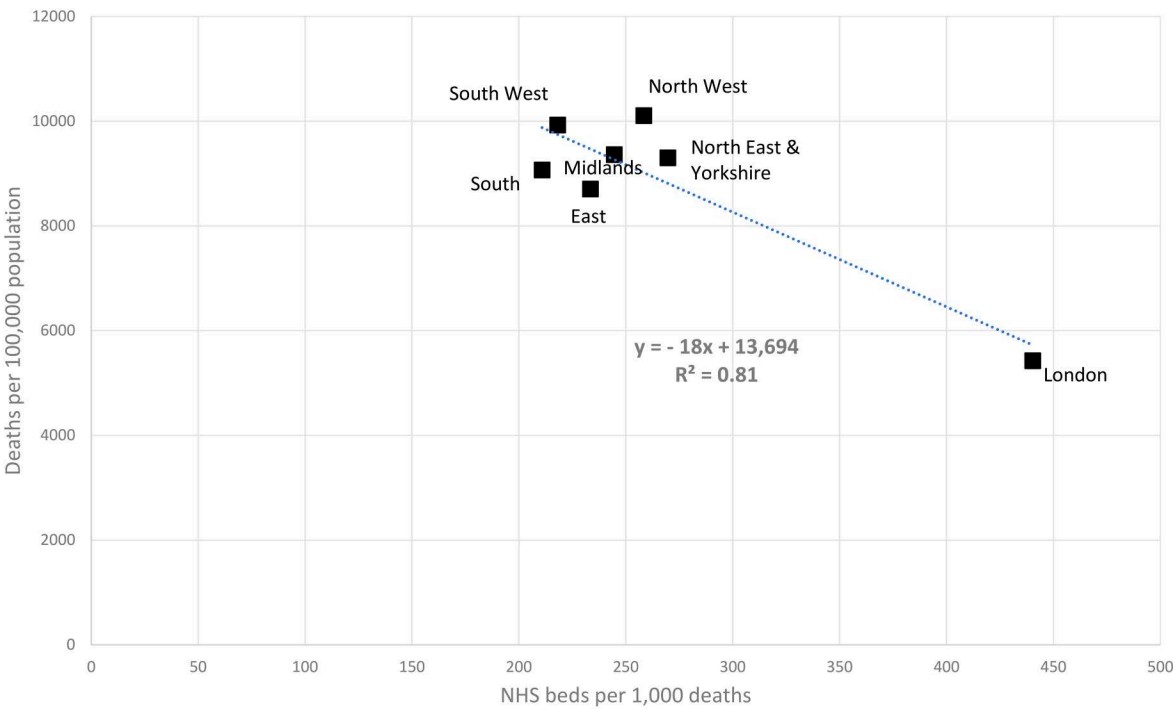

**Fig 6. The number of NHS inpatient beds per 1,000 deaths in 2019/2020 plotted against the mortality rate per 100,000 total population.** Data are shown for seven regions in England. Linear trend and the regression equation with $R^2$ is also shown.

Within the 60 consecutive years of reduction there were two periods of accelerated bed reductions, the first starting in the 1980s and the second starting in the 2000s. As a result, nearly 80% of NHS inpatient bed capacity was taken out. If instead of these periods of accelerated bed reductions there has been a continuation of the rate of reduction prior to the 1980s, then we estimate there would have been closer to a 60% reduction in NHS bed capacity. If this had occurred, then at the start of the COVID-19 pandemic England would have had a rate of bed provision that would have been in the middle of the G7 nations. However, this was not the case, and the UK entered the pandemic with the lowest rate of inpatient bed provision amongst the G7 group of nations. We believe that this very low rate of bed provision in the UK is one of the primary reasons why the NHS has faced repeated winter bed pressures and why a decision was made about the need to "protect the NHS" during the Covid pandemic.

Evidence to support this is that the rate of the increase in the ratio of deaths to Acute and General beds had started to accelerate in the years prior to the pandemic. Life expectancy in England increased during the first five decades of the study period. It then plateaued around the same time as the ratio of deaths to beds went above 4, and life expectancy started to fall at the same time as the ratio of deaths to beds approached 5. Data suggest that this ratio jumped above 5 as the number of deaths during the pandemic years exceeded the numbers reported in this paper. This reflected the concern expressed in the slogan "protect the NHS" as there was a fear that this ratio would become so high as to overwhelm the NHS's ability to keep admitting critically ill patients.

This fundamental problem remains unresolved and will reoccur if a further pandemic occurs. This will be most evident if this is again a disease that causes respiratory problems or particularly affects the elderly, the mentally ill and those with learning disabilities. Furthermore, a strategy of meeting the funding gap faced by the NHS by closing inpatient beds is not sustainable in the long term and is not compatible with preparing for future winter bed pressures or pandemics. Any further cuts in beds will make addressing the current NHS waiting lists more difficult and this may further exacerbate existing health inequalities.

Although data on length of stay was only available from 1998, our results show that most specialities accessing General & Acute beds showed an increase in admissions whilst at the same time showing a reduction in Occupied Bed Days. This was due to much shorter lengths of stay, indicating that part of the explanation for the reduction in beds is improvements in the efficiency and effectiveness of care, in part related to new procedures and treatments. Furthermore, the expansion of Day Only beds means that some of these will have been used for procedures that previously required overnight admission.

The pattern in Mental Illness and Learning Disability services was quite different with marked reduction in admission occurring at the same time as increased length of stay. Both of these support the widely held opinion that the threshold for admission has steadily risen and is currently very high with only the most severely ill or those with the highest risk meeting the threshold for admission. Maternity services showed a modest reduction in admissions but a doubling in the ratio of births to maternity beds with mothers spending one day in hospital to give birth instead of two days.

We also demonstrate that by the end of the study period there was significant regional variation in NHS inpatients services. London continues to stand out with regard to inpatient bed provision when compared with other regions particularly when mortality rates are considered. There are likely to be a number of reasons for this including the age profile of the population, and the location of many tertiary hospital services in London. However, it also raises the possibility that the lower mortality rate in London may in part be explained by the higher relative provision of NHS inpatient beds. These results throw some light on equity of access to inpatient care across England.

Furthermore, our results demonstrate that the NHS is no longer providing the same level of inpatient service in every region of the country. The greatest variations in bed provision between regions were in bed categories that experience the largest reductions in the preceding 60 years. The reduction and variation in learning disability services stands out in both regards: there was a 98% reduction in Learning Disability inpatient bed provision over 60 years and a twelve-fold variation

between regions in provision of NHS Learning Disability inpatient beds in 2019/20. The second highest reduction was in Mental Illness beds, and these showed the second greatest level of variation between regions – nearly two-fold.

In addition, those specialities that experienced the greatest reductions in NHS inpatient episodes of care were the same specialities that showed the highest involvement of the Independent Sector in the provision of NHS funded inpatient care at the end of the study. There were three broad groupings in this regard. Psychiatric services had the greatest reduction in episodes and had the highest provision by the Independent Sector. General Medicine and related specialities experienced an increase in inpatient episodes and by the end of the study had minimal involvement of the Independent Sector. Surgical specialities and Obstetric & Gynaecology fell somewhere between these two extremes. One interpretation of these findings is that the NHS highly values specialities such as General Medicine, Paediatrics, Cardiology and Oncology and has sought to keep these services within the NHS. In contrast it does not place the same value on Psychiatric services or does not view the provision of such services as its core business. In contrast the Independent Sector has been willing to fill the gap created.

If certain regions decide to use the Independent Sector to provide types of inpatient care then policies should be developed stating where these services should be located: within the region or at distance that is accessible for patients and their families and carers; and how best they can be integrated with the wider regional health network in terms of digital and other electronic systems. The impact of these changes should also be evaluated given concerns about impact on outcomes [23].

The first period of accelerated bed closures that started in the 1980s accounted for the largest number of bed reductions and affected the whole NHS albeit to different degrees according to bed category. It started in a period of severe economic recession in the early 1980s and in the middle of this phase the UK government passed the NHS Community Care Act (1990) [24]. This Act focused on moving patient care from hospital to community settings for all specialities. More recently this the NHS Long Term Plan reiterates this shift by aiming "to provide fast support to people in their own homes as an alternative to hospitalisation." This applies to both physical and mental health problems. This process of moving to "care in the community" started earlier in mental health and learning disability services. In 1975 the Department of Health published "Better services for the mentally ill" [25] which deliberately set out the intention to move away from institutional care and develop community services to replace large hospitals and asylums. Furthermore, across all services there has been a sustained drive to reduce length of stay by identifying bottle necks, mapping the discharge process, setting discharge dates and actively managing the discharge process. In addition, there has been a focus on patients with long lengths of stay particularly those in mental illness or learning disability units to determine if they can be looked after in the community.

The second period of accelerated bed closures followed on from the publication of the NHS Plan in 2000 that aimed to increase bed numbers and staff, and improve inpatient facilities by aiming to build 100 new hospitals using the Private Finance Initiative, and encouraging public-private partnerships in the delivery of NHS funded care. However, we demonstrate that a second period of accelerated NHS bed closures followed.

The reduction in Geriatric bed capacity by three quarters, and even more if age is controlled for, is striking. This occurred as life expectancy increased and the number of old and very old people has increased markedly. Whilst the population of England increased by 30% during the study, the number aged 65 years and older doubled, and the number aged 80 years and older increased more than 300% from just under 2% of the population in 1960 to 5% in 2019. The limited capacity of the NHS to admit and treat the elderly and frail in hospital has become apparent each winter in recent years [26] and was made more apparent during the COVID-19 pandemic to which the elderly are particularly at risk [27]. We believe the impact of the closure of Geriatric beds identified in this paper will need to be considered when plans are drawn up in preparation for any future pandemics particularly those that may disproportionately affect the elderly and the respiratory system requiring supplemental oxygen which most nursing homes are not able to provide consistently.

The high rate of closure of Mental Illness beds occurred right from the start of the study period in the 1960s and fits with the policy articulated by Enoch Powell's speech in 1961 [28]. Interestingly, the same did not apply to Learning Disability beds. Initially these showed low rates of bed closure until the 1980s when there was a dramatic change to very high rates of bed closure which overtook the rate of closure of Mental Illness beds and continued right up to the end of the study period. A similar pattern was seen for Geriatric beds and Maternity beds with low rates of closure in the 1960s and 1970s changing to high rates of bed closure from 1984–1997/8. These dramatic changes in the rate of bed closures in such different and divergent services does not appear to reflect any change in clinical practice but rather changes in policy including the shift from hospital to community. Elderly patients with dementia, and younger adults with enduring mental illness and intellectual disabilities are considered some of the most vulnerable in society and they are also at increased risk from COVID-19 [29–31]. It is striking that these are the inpatient clinical areas that the NHS has divested itself of and the private sector was commissioned to provide care for. One of the questions raised by this paper is why were these areas chosen rather than the Acute sector and the Maternity sector? This warrants further research.

This extreme variation in Learning Disability inpatient bed provision is worthy of note and requires further research. The understanding and approaches to supporting people with Learning Disability has evolved significantly over the past 60 years and could explain a significant proportion of the bed reductions seen here over the time period. In 1993 The Mansell report [32], which came nearly 10 years after the onset of the accelerated period of bed reduction, placed emphasis on support in the community replacing hospitalisation for Learning Disability patients. More recently, following the Winterbourne View scandal, Transforming Care agenda has sought to further reduce bed numbers whilst aspiring to bolster community provision [33,34].These drivers to reduce Learning Disability bed numbers have also been incorporated into consecutive English policies to move from hospital based services to community delivered care such as the Five Year Forward View in 2014, the 2019 NHS Long Term Plan, and most recently in the 2025 "Fit for the Future: 10 year plan for England" [35–37].

However, there are concerns that the reduction in NHS beds including secure beds is having a negative impact on patients and limiting the ability of courts and prisons to transfer mentally disordered offenders who require treatment in hospital, as well as a stalling in the growth of community provision [38]. This situation has been further exacerbated by the uncovering of more cases of abuse at hospitals, resulting in further bed reductions, and consequent increased use of non-specialist Mental Illness beds when admission is required. The regional differences we report in Learning Disability services indicates that the NHS no longer provides the same level of service for this very vulnerable group of patients in different regions.

Sixty decades of continuous, yearly reductions in NHS hospital beds has the left inpatient hospital provision vulnerable in England. This vulnerability is exposed most winters, but was particularly apparent during the COVID-19 pandemic. Future planning will need to address this and in particular to consider the needs of the elderly and those with mental illness and learning disability, as these groups have been most affected by NHS bed closures. The policy of moving the care of geriatric patients from hospitals to nursing homes made the NHS particularly vulnerable to a pandemic that affected the respiratory system, as supplementary oxygen as is not reliably available in nursing homes. Nor is it available in mental illness beds or learning disability beds even though patients on these units are frequently being treated for co-morbid physical health issues. Future research should compare the changes in Scotland, Wales and Northern Ireland to those that occurred in England. The provision of NHS funded private sector inpatient services should be publicly available at a regional level to see if this mitigates the significant regional differences identified in this paper.

## Strengths and limitations

Due to the nature of the analysis undertaken this paper only identifies associations and we have not demonstrated causation. The extended period studied is a strength and resulted in a more comprehensive description of the pattern of closures of five categories of NHS beds. The 60-year period can also be considered a limitation due to the different economic cycles and several changes in NHS regulation that occurred during this time period. The use of Hospital Episode

(HES) Statistics in addition to Hospital Activity Statistics is another strength as the former measures inpatient activity whilst the later measures inpatient bed numbers. In addition, from 1998/99 demographic information is available including age and gender for HES.

A limitation is the lack of information linking HES data to socioeconomic measures for all but the most recent years. For example, data on the amount of inpatient activity according to levels of deprivation was only available for the last three years of the 60-year period and was not available for any year for hospital bed numbers. Nor did we have access to data linking HES activity to population density, chronic disease prevalence, or measures of co-morbidity.

Including all NHS inpatient beds and the five broad categories is another strength. In 2010 the NHS stopped reporting on Geriatric beds as separate from Acute beds and combined these into a General and Acute category. This was due to it being difficult to clearly demarcate the two with acute beds being used for care of the elderly and vice versa. However, this is not an issue that is unique to Acute and Geriatric beds. With the severe shortage of Learning Disability beds it is common for patients with intellectual disabilities to be admitted into Mental Illness beds. In addition, it is common for patients with mental illnesses and intellectual disabilities to occupy Acute and Geriatric beds whilst waiting for a Mental Illness bed to become available or whilst being treated for a physical health manifestation of their mental disorder.

The inclusion of data only from NHS England is a limitation. Extending this research to include Wales, Scotland and Northern Ireland would improve the robustness and generalisability of these findings. Extending the research to include health systems in other comparable economies would enable meaningful comparisons to be made. Detailed data on bed occupancy was not available for the whole study period and is another limitation. Bed numbers were not available for the regions of England for the whole time period, and even for those years that bed data was available comparisons between years were difficult to make due to the reorganisation of the health regions during study period.

The lack of comprehensive information on the number of inpatient hospital beds in private hospitals in England is a limitation. Broadly there are two types of private inpatient hospital care in England. The first is funded by individuals or insurance schemes, the second is funded by the NHS but provided by the private sector. Data on the number of inpatient beds provided by the independent sector for the period covered were not available. Greater transparency with comprehensive data submission by the Independent Sector being a requirement, would enable future research to analysis the optimal combination of NHS and Independent Sector in the provision of state funded health care.

## Conclusions

The NHS experienced continuous reductions in inpatient bed capacity over 60 years with two periods of accelerated reductions in capacity, the first in the 1980s and the second in the 2000s. These reductions and associated changes in inpatient activity have differed according to medical speciality with Psychiatric services affected to the greatest extent, General Medicine and related specialities affected the least, and General Surgery and related specialities including Obstetrics and Gynaecology somewhere in between. By the end of the study the NHS inpatient provision had changed dramatically and the NHS no longer directly provided the same level of inpatients services in the different regions of England. At the end of the study the Independent Sector was providing a significant proportion of NHS funded care particularly in those specialities that had experienced the greatest reduction in NHS inpatient beds.

## Supporting information

**S1 File. Supplementary data file containing NHS inpatient bed numbers between 1960 and 2019/20.** All data used in this analysis is available here.
(XLSX)

**S2 File. Supplementary results.**
(DOCX)

## Author contributions

**Conceptualization:** Patrick Keown, Ross Alder, Georgina Wild, Iain Mckinnon.

**Data curation:** Patrick Keown, Ross Alder, Georgina Wild.

**Formal analysis:** Patrick Keown, Ross Alder, Georgina Wild, Scott Weich, Luke Ouma, Iain Mckinnon.

**Methodology:** Patrick Keown, Ross Alder, Georgina Wild, Luke Ouma.

**Project administration:** Ross Alder, Georgina Wild.

**Supervision:** Patrick Keown, Iain Mckinnon.

**Writing – original draft:** Ross Alder, Georgina Wild.

**Writing – review & editing:** Patrick Keown, Scott Weich, Luke Ouma, Iain Mckinnon.

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
