## [Decision Letter · Decision Letter 0]

18 Jun 2024

PONE-D-24-12261Long term trends in NHS inpatient bed provision in England, 1960-2020PLOS ONE

Dear Dr. Weich,

Thank you for submitting your manuscript to PLOS ONE. After careful consideration, we feel that it has merit but does not fully meet PLOS ONE’s publication criteria as it currently stands. Therefore, we invite you to submit a revised version of the manuscript that addresses the points raised during the review process.

We look forward to receiving your revised manuscript.

Kind regards,

Ben Green

Academic Editor

PLOS ONE

2. We note that your Data Availability Statement is currently as follows: [All relevant data are within the manuscript and its supporting information files.]

Additional Editor Comments:

Thank you for your original and interesting paper which we would like to accept but with some necessary revisions, suggested by reviewers:

1. Please can you specifically address in the introduction or conclusion regarding psychiatric bed provision and its decline referencing:

Hospital admission and community treatment of mental disorders in England from

1998 2012 (Green and Griffiths, 2014 ) [Gen Hosp Psychiatry2014 Jul-Aug;36(4):442-8. doi: 10.1016/j.genhosppsych.2014.02.006.Epub 2014 Feb 12.]

2. The aim of the paper is to provide context to understand why it was deemed necessary to “protect the NHS”. For readers outside the UK who may not be familiar with this concept, it's important to be clear what “protect the NHS” meant.

And

Page 5: Clarify what is meant by intermediate care beds and residential care beds. Additionally, on page 9 – “2010/11 - change in recording process for NHS inpatient beds”. How does this affect our interpretation of the data after 2010/11?

3. Explain if there were other categories of beds considered and why these specific five were chosen.

4. Page 7: What are AIC/BIC? Clarify.

5. Typo, Page 7: “It other categories” – “In other categories”

6. Page 7: You say that the size of the reduction of inpatient beds was 319,074 but numbers of inpatient beds went from 447,457 in 1960 to 128,277 in 2019/20, which equals 319,180 not 319,074.

7. Suggested change, Page 8: “The largest numbers of beds were reduced in” – “The largest reduction in number of beds was in”

8. Typo, Page 9: “1980s.this” – “1980s. This”

9. Typo, Page 10: “1960s and 1970ss”

10. Typo, Page 11: consider rewording “NHS inpatient bed provision in London showed a distinct pattern because with the highest rates of both Mental illness bed provision…and Maternity bed provision”

11. Typo, Page 17: “Northern Island” to “Northern Ireland”

12. Page 10: “no single year when there was an increase in NHS inpatient bed provision” Is this overall? Because you say on page 9 that “Maternity beds…had an increase in bed provision in several years in the 1960s”. Clarify.

13. Page 14 – “strategy of meeting the funding gap faced by the NHS by closing inpatient beds is not sustainable in the long term”. This is a bold, yet justifiable statement. I would like to see this elaborated more. What are the direct implications of bed reductions for the NHS overall, not just during an epidemic? “Protect the NHS” brought to light the issues of bed reductions and it’s unlikely that the NHS will use what we learned from COVID-19 to prepare for the next pandemic. What about the now? Direct implications of bed reductions could include: impact on patient care and outcomes (e.g., increased waiting times, overcrowding in remaining facilities, displacement of patients), pressure on community and outpatient services (e.g., shift to community care and rehospitalization rates), financial and operational challenges (e.g., short-term savings vs long-term costs and resource allocation), healthcare workforce implications (e.g., staffing levels and morale), and equity and access to care (e.g., regional disparities highlights health inequalities).

14. Page 14 – “reduction in Geriatric bed capacity by three quarters, and even more if age is controlled for”. Where is this data?

15. Page 16 “These dramatic changes in the rate of bed closures in such different and divergent services does not appear to reflect any change in clinical practice but rather changes in policy including the shift from hospital to community”. For the Mental Illness and Learning Disability categories, you have clearly shown that “care in the community” was a major policy change that affected bed reductions. What about for the other categories?

16. Other things to consider:

a. Over the years, the understanding and approach to treating and supporting individuals with learning disabilities have evolved significantly. How could these changes have had a direct impact on the provision and reduction of inpatient beds for this population? Would this explain all of the bed reduction?

b. Over the past several decades, birth rates in many developed countries, including England, have undergone significant changes. How might these changes directly impact the demand for and overall reduction in inpatient beds? Would this explain all of the bed reduction?

17. The authors state that there are two main limitations of the study: NHS England data only and lack of information on number of inpatient hospital beds. However, the authors only publish average number of hospital beds per year per category, which raises the question of what additional data could have been controlled for to enhance the study’s robustness. Did the authors not have access to additional data? If not, please state. Otherwise, important variables should be address, such as: socioeconomic factors, population density, urbanization, chronic disease prevalence, and healthcare access (e.g., number of healthcare providers per capita and availability of outpatient services). The absence of these additional data points is a significant limitation for several reasons:

a. Incomplete understanding of influences to changes in bed numbers: Without controlling for a broader range of variables, the study may not fully capture the factors driving changes in bed numbers. This can lead to an incomplete or biased understanding of the trends observed.

b. Reduced accuracy of findings: The lack of detailed control variables can reduce the accuracy and reliability of the study's findings. For instance, attributing bed reductions solely to policy decisions without considering socioeconomic or demographic factors could oversimplify complex issues.

c. Limited Policy Relevance: Policymakers rely on comprehensive data to make informed decisions. Studies that do not consider all relevant variables may provide less actionable insights, limiting their usefulness for crafting effective healthcare policies.

d. Potential Confounding Factors: Uncontrolled confounding factors could distort the relationship between the variables studied and the outcomes observed. This could lead to incorrect conclusions about the causes and effects of bed reductions.

An additional comment was regarding the literature research supporting this manuscript, which was considered non exhaustive, and was missing several relevant works. Could you please ensure that you consider all relevant literature for the interpretation of your data? Extensive comparison between England and other countries has been conducted on this basis including total beds and critical care beds. Perhaps consider modifying this approach to show if the reduction in bed numbers is justified. This approach would be highly beneficial in terms of the regional bed numbers comparison. In terms of maternity it would be useful to show the ratio of available beds per birth. Births being the fundamental driver for demand of this type of beds. For some of the other bed types a proxy population profile could be constructed. It may also be useful to show how average length of stay has changed over the years and to what extent this may have contributed to reduced bed demand. Another suggestion could be a short review of the factors regulating bed demand and consequent bed modeling. Research has shown that bed numbers in Japan are over stated due to the incorrect inclusion of nursing home beds in the curative bed total. Some comment on the trends in bed occupancy may also be helpful.

Reviewers' comments:

Reviewer's Responses to Questions

**Comments to the Author**

1. Is the manuscript technically sound, and do the data support the conclusions?

Reviewer #1: Partly

Reviewer #2: No

Reviewer #3: Partly

2. Has the statistical analysis been performed appropriately and rigorously? 

Reviewer #1: Yes

Reviewer #2: No

Reviewer #3: Yes

3. Have the authors made all data underlying the findings in their manuscript fully available?

Reviewer #1: Yes

Reviewer #2: Yes

Reviewer #3: Yes

4. Is the manuscript presented in an intelligible fashion and written in standard English?

Reviewer #1: Yes

Reviewer #2: Yes

Reviewer #3: Yes

5. Review Comments to the Author

Reviewer #1: Dear Authors,

This is an interesting piece which the readers will find relevant. However, it lacks any perspective as to whether the fundamental demand for beds has changed. A recent review has suggested that bed numbers may be better understood in the light of the number of deaths per 1000 population. Extensive comparison between England and other countries has been conducted on this basis including total beds and critical care beds. May I suggest that you modify this approach to show if the reduction in bed numbers is justified. This approach would be highly beneficial in terms of the regional bed numbers comparison. In terms of maternity it would be useful to show the ratio of available beds per birth. Births being the fundamental driver for demand of this type of beds. For some of the other bed types a proxy population profile could be constructed. It may also be useful to show how average length of stay has changed over the years and to what extent this may have contributed to reduced bed demand. Another suggestion could be a short review of the factors regulating bed demand and consequent bed modeling. My own research has shown that bed numbers in Japan are over stated due to the incorrect inclusion of nursing home beds in the curative bed total. Some comment on the trends in bed occupancy may also be helpful.

This is a useful study which can be made better by some additional research both in the literature and the supporting analysis.

Reviewer #2: Thank you for the opportunity to review this manuscript. It is interesting to see that over the past 60 years, population-adjusted NHS bed provision has decreased at such a large proportion. This phenomenon has not been unique to the UK but rather common among most high-income countries. However, the reason behind the observation should be examined more closely with broader considerations for potential contributing factors. For instance, reduction in hospital bed capacity could reflect a degradation in quality of care but could also be the result of increased efficiency. Therefore, one might wonder whether the outcomes measured could reflect the changes in the quality of public health services, advances in medicine, efficiency of hospital care, and simply, evolutions in standard of care.

Given healthcare utilization is heavily skewed towards the end of life, the nearness to death effect cannot be neglected when looking at number of hospital beds. One commonly used method is to evaluate the logarithmic relationship between beds per 1000 deaths and deaths per 1000 population (in this case, per 100,000 populations) to take population age structure into consideration. The nearness to death effect is ultimately important for policy making and resource allocation for future hospital beds.

Despite the limitations of evaluating changes in population-adjusted hospital beds, it is crucial to evaluate the current situation and use it to inform future policy making. The study can be greatly more informative if the authors could provide insights on comparison between healthcare utilization between the public and private sectors in the UK, and future trends despite not having the exact number of bed in the private sector.

Reviewer #3: The manuscript titled "Long term trends in NHS inpatient bed provision in England, 1960-2020" examines the reduction of NHS inpatient hospital beds in England from 1960-2020, focusing on 5 categories: Acute, Geriatric, Maternity, Mental illness, and Learning Disability. The study aims to quantify the change in the total number and different categories of NHS inpatient hospital beds within the time period, and to provide context to understand why it was deemed necessary to “protect the NHS”.

The study provides valuable historical insight into the reduction of NHS inpatient hospital beds over a significant period (1960-2020). By examining five key categories, the research highlights critical trends and alludes to shifts in healthcare resource allocation. Understanding these changes is crucial for policymakers to make informed decisions about future healthcare planning and resource management, especially in light of ongoing debates about the sustainability and efficiency of the NHS.

By measuring regional differences in bed reductions at the end of the study period, the research sheds light on healthcare equity across England. Identifying regional disparities can inform targeted interventions to ensure that all regions have adequate healthcare resources. This aspect of the study is particularly significant as it can drive policies aimed at reducing inequalities and improving access to healthcare services nationwide. This was not discussed in depth in the paper, and it should be.

I commend the authors for undertaking this important study, providing crucial insights. While I believe this paper is a significant contribution to the field, and I agree with the overall message conveyed, there are several areas where its strength could be further enhanced, especially in the Discussion section. Please see below:

1. The aim of the paper is to provide context to understand why it was deemed necessary to “protect the NHS”. For readers outside the UK who may not be familiar with this concept, it's important to be clear what “protect the NHS” meant.

2. Page 5: Clarify what is meant by intermediate care beds and residential care beds. Additionally, on page 9 – “2010/11 - change in recording process for NHS inpatient beds”. How does this affect our interpretation of the data after 2010/11?

3. Explain if there were other categories of beds considered and why these specific five were chosen.

4. Page 7: What are AIC/BIC? Clarify.

5. Typo, Page 7: “It other categories” – “In other categories”

6. Page 7: You say that the size of the reduction of inpatient beds was 319,074 but numbers of inpatient beds went from 447,457 in 1960 to 128,277 in 2019/20, which equals 319,180 not 319,074.

7. Suggested change, Page 8: “The largest numbers of beds were reduced in” – “The largest reduction in number of beds was in”

8. Typo, Page 9: “1980s.this” – “1980s. This”

9. Typo, Page 10: “1960s and 1970ss”

10. Typo, Page 11: consider rewording “NHS inpatient bed provision in London showed a distinct pattern because with the highest rates of both Mental illness bed provision…and Maternity bed provision”

11. Typo, Page 17: “Northern Island” to “Northern Ireland”

12. Page 10: “no single year when there was an increase in NHS inpatient bed provision” Is this overall? Because you say on page 9 that “Maternity beds…had an increase in bed provision in several years in the 1960s”. Clarify.

13. Page 14 – “strategy of meeting the funding gap faced by the NHS by closing inpatient beds is not sustainable in the long term”. This is a bold, yet justifiable statement. I would like to see this elaborated more. What are the direct implications of bed reductions for the NHS overall, not just during an epidemic? “Protect the NHS” brought to light the issues of bed reductions and it’s unlikely that the NHS will use what we learned from COVID-19 to prepare for the next pandemic. What about the now? Direct implications of bed reductions could include: impact on patient care and outcomes (e.g., increased waiting times, overcrowding in remaining facilities, displacement of patients), pressure on community and outpatient services (e.g., shift to community care and rehospitalization rates), financial and operational challenges (e.g., short-term savings vs long-term costs and resource allocation), healthcare workforce implications (e.g., staffing levels and morale), and equity and access to care (e.g., regional disparities highlights health inequalities).

14. Page 14 – “reduction in Geriatric bed capacity by three quarters, and even more if age is controlled for”. Where is this data?

15. Page 16 “These dramatic changes in the rate of bed closures in such different and divergent services does not appear to reflect any change in clinical practice but rather changes in policy including the shift from hospital to community”. For the Mental Illness and Learning Disability categories, you have clearly shown that “care in the community” was a major policy change that affected bed reductions. What about for the other categories?

16. Other things to consider:

a. Over the years, the understanding and approach to treating and supporting individuals with learning disabilities have evolved significantly. How could these changes have had a direct impact on the provision and reduction of inpatient beds for this population? Would this explain all of the bed reduction?

b. Over the past several decades, birth rates in many developed countries, including England, have undergone significant changes. How might these changes directly impact the demand for and overall reduction in inpatient beds? Would this explain all of the bed reduction?

17. The authors state that there are two main limitations of the study: NHS England data only and lack of information on number of inpatient hospital beds. However, the authors only publish average number of hospital beds per year per category, which raises the question of what additional data could have been controlled for to enhance the study’s robustness. Did the authors not have access to additional data? If not, please state. Otherwise, important variables should be address, such as: socioeconomic factors, population density, urbanization, chronic disease prevalence, and healthcare access (e.g., number of healthcare providers per capita and availability of outpatient services). The absence of these additional data points is a significant limitation for several reasons:

a. Incomplete understanding of influences to changes in bed numbers: Without controlling for a broader range of variables, the study may not fully capture the factors driving changes in bed numbers. This can lead to an incomplete or biased understanding of the trends observed.

b. Reduced accuracy of findings: The lack of detailed control variables can reduce the accuracy and reliability of the study's findings. For instance, attributing bed reductions solely to policy decisions without considering socioeconomic or demographic factors could oversimplify complex issues.

c. Limited Policy Relevance: Policymakers rely on comprehensive data to make informed decisions. Studies that do not consider all relevant variables may provide less actionable insights, limiting their usefulness for crafting effective healthcare policies.

d. Potential Confounding Factors: Uncontrolled confounding factors could distort the relationship between the variables studied and the outcomes observed. This could lead to incorrect conclusions about the causes and effects of bed reductions.

6. PLOS authors have the option to publish the peer review history of their article (what does this mean? ). If published, this will include your full peer review and any attached files.

**Do you want your identity to be public for this peer review?** For information about this choice, including consent withdrawal, please see our Privacy Policy .

Reviewer #1: **Yes: ** Dr Rodney P Jones

Reviewer #2: No

Reviewer #3: No

---

## [Decision Letter · Decision Letter 1]

3 Mar 2025

PONE-D-24-12261R1Long term trends in NHS inpatient bed provision in England, 1960-2020PLOS ONE

Dear Dr. Weich,

Thank you for submitting your manuscript to PLOS ONE. After careful consideration, we feel that it has merit but does not fully meet PLOS ONE’s publication criteria as it currently stands. Therefore, we invite you to submit a revised version of the manuscript that addresses the points raised during the review process.

We look forward to receiving your revised manuscript.

Kind regards,

Ben Green

Academic Editor

PLOS ONE

Journal Requirements:

Additional Editor Comments :

Following revision one reviewer has suggested the following points be addressed please:

1. Comment 14 (Page 14 – “reduction in Geriatric bed capacity by three quarters, and even more if age is controlled for”. Where is this data?). The authors answered, “This is shown in Appendix 1.”.

• Is Appendix 1 Supplementary file 1? This is a Figure. Maybe they referred to Supplementary file 2 where there is a table showing “The number of five different categories of NHS beds and the provision of beds per 100,000 population in England in 1960 and 2019/20.” However, nothing is mentioned about controlling for age besides commenting that Geriatric figures only include 65+ patients and the other categories total population.

2. Comment 16a (Over the years, the understanding and approach to treating and supporting individuals with learning disabilities have evolved significantly. How could these changes have had a direct impact on the provision and reduction of inpatient beds for this population? Would this explain all the bed reduction?). The authors replied, “The paragraph at the end of page 9 addresses this point and we have added the following: “The understanding and approaches to supporting people with Learning Disability has evolved significantly over the past 60 years, and could explain a significant proportion of the bed reductions seen here over the time period.””.

• I think the authors should make these comments if they are supported by the literature. Otherwise, it looks authors’ speculation with no evidence backup. I suggest that the authors include evidence that supports this. This can be explained by changes in the health policy by offering more services at the community level such as in the NHS Five Year Forward Review or the NHS Long Term Plan.

3. Comment 17 (The authors state that there are two main limitations of the study: NHS England data only and lack of information on number of inpatient hospital beds. However, the authors only publish average number of hospital beds per year per category, which raises the question of what additional data could have been controlled for to enhance the study’s robustness. Did the authors not have access to additional data? If not, please state. Otherwise, important variables should be address, such as: socioeconomic factors, population density, urbanization, chronic disease prevalence, and healthcare access (e.g., number of healthcare providers per capita and availability of outpatient services) (…)). The authors responded “Data on Gross Domestic Product and annual change in this has been added to the analysis as well as the the number of deaths and births. We have also added data on Day Only beds, length of stay, the number of inpatient episodes of care, occupied bed days from 1998 onwards to help address the points raised. Reference has been added along with “including psychiatric beds” added to the introduction. The reference section has also been reviewed as suggested with five addition references added and some removed.”.

• I don´t think the authors incorporated the comments suggested in previous reviews. So far, the study is very descriptive with little information to understand the main drivers of this reduction in the number of beds. It could be understood that there was a disparity in the population growth and the number of available beds, but this is unclear. If the authors are using HES data, they can provide more information controlling for socioeconomic factors, for example. Also for Charlson co-morbidities. The authors should think about this question and answer the question addressed by reviewers.

Additional comments:

4. The document provided includes a total of 31 pages. But from page 15, page numbers do not continue previous numbering. Can the authors revise this issue, please?

5. There is a typo on page 3. It says COID19 and should say COVID-19.

6. On page 3 the authors should mention why the ratio of deaths to inpatient bed is important for international comparisons.

7. Page 4 – Material and Methods – The authors mention that they included Day Only beds. Could you explain what kinds of procedures include this concept? I think they are linked to outpatient care (OPC) procedures, but I think this could be clarified, for example if you need to be admitted the day before or you can undergo the procedure in the same day and you will be discharge in few hours during the day.

8. Page 5 – The authors mentioned on page 4 that they got the numbers of inpatient beds following a data request. However, they mention on page 5 that their main data source is HES. Can you clarify this? Also, which HES module you asked for APC, OPC, ACC,…? You can briefly describe each module in 1-2 sentences.

9. Page 5 – how many hospitals are you including? In England there are more than 200 NHS hospitals and some private providers. Since you are also discussing the participation of the private (independent) sector, it would be good to know the proportion in hospitals in England.

10. Page 6 – In the Statistical analysis section, the authors mention “The annual percentage change (APC) in bed provision was the variable used to measure rate of change”. Is this relative to population change or relative change to previous year?

11. Page 7 – RCS & LOESS methods –These methods only provide information on potential associations between variables, not causal effects. Therefore, the authors need this to be clear. Moreover, it would be good to include a Supplementary material where the authors explain briefly how the RCS operates (i.e. using Lusa L, Ahlin Č. Restricted cubic splines for modelling periodic data. PloS One. 2020 Oct 28;15(10):e0241364. Doi: 10.1371/journal.pone.0241364). For the LOESS methods, I suggest the authors clarify this is a function that smooths fitted values.

12. Page 8 – there is a typo in the 3rd paragraph. Where it say “The largest reductions in numbers…”, then it should say “The largest reduction in the number of beds”

13. Page 8 – I suggest the authors move the paragraph above Day Only beds on page 9 to page 8 above the paragraph starting “At the start of the study Acute beds and Mental Illness beds (…)”.

14. Page 9 – regarding Geriatric beds, what is the proportion of 65+ patients in the population and the sample?

15. Page 11 – Figure 3 shows declining and accelerating periods. The accelerating bed closures in 1980-1990 was briefly explained in the Discussion (page 7 +15). What about the second period 2000-2010? I am sure there were changes in the provision of the service at community level but also health budget cuts that may contribute to this decision. Could you compare bed closures to population growth in those years?

16. Page 12 – last paragraph. Please include a gap when you say, “11 million”. Are you saying that the 11% reduction in Occupied beds is explained by halving the average lengths of admissions? I think you wanted to say length of stay of admitted patients. Is this explained by new procedures/treatments or less severe diagnoses?

17. Page 14 – what is the proportion of beds provided from the Independent (private) sector compared to the NHS provision?

18. Page 1 (+15) – From Table 2 onwards, the authors should renumber continuing from page 15.

19. Page 2 (+15) – Table 4 includes p-values. It is suggested that the table incorporates standard errors. The Table incorporates p-values and highlights coefficients that are statistically significant from the Poisson regression. Which variables were included in the regression model? Moreover, the authors may consider replacing p-values with stars, indicating the significance level in the coefficients (*0.05 significance level, **0.01 and ***0.001). That will simplify the understanding of the Table. Did the authors control for providers fixed effects to account for heterogeneity?

20. Page 3 (+15) – There is a typo in the first paragraph. Where it says, “This ratio was much higher in London (440, N=1) then…”. This “then” should be replaced with “than”.

21. Page 7 (+15) – As I mentioned in Comment 15, please discuss the second period of accelerated bed closures 2000-2010.

22. Page 11 (+15) – Strengths and limitations. I am not sure if a 60-year period is a strength. In such long period there is too much heterogeneity caused by different economic cycles and changes in regulation. I would say that using HES data is a strength if the authors justify this rich data with analyses incorporating demographics and other relevant socioeconomic controls. But the authors should make clear if they are using HES in their analysis. So far, it is not clear.

Reviewers' comments:

Reviewer's Responses to Questions

**Comments to the Author**

1. If the authors have adequately addressed your comments raised in a previous round of review and you feel that this manuscript is now acceptable for publication, you may indicate that here to bypass the “Comments to the Author” section, enter your conflict of interest statement in the “Confidential to Editor” section, and submit your "Accept" recommendation.

Reviewer #4: (No Response)

Reviewer #5: (No Response)

2. Is the manuscript technically sound, and do the data support the conclusions?

Reviewer #4: Partly

Reviewer #5: Partly

3. Has the statistical analysis been performed appropriately and rigorously? 

Reviewer #4: Yes

Reviewer #5: No

4. Have the authors made all data underlying the findings in their manuscript fully available?

Reviewer #4: Yes

Reviewer #5: No

5. Is the manuscript presented in an intelligible fashion and written in standard English?

Reviewer #4: Yes

Reviewer #5: Yes

6. Review Comments to the Author

Reviewer #4: This paper examines trends in inpatient beds in the UK from 1960 to 2020. This is the first revision of the manuscript, but I can’t see the “responses to reviewers”. However, based on the revised and track-changed version of the manuscripts, my comments/questions for the paper are:

1. What is the rationale/hypothesis to test for a trend in NHS inpatient bed provision in England in 1960-2010? (e.g., technological progress, policy changes, demographic shifts)?

2. What are the results of detailed analysis (e.g., restricted cubic splines, fractional polynomials, LOESS?

3. What variables did the authors use to control for factors affecting the trend (e.g., changes in reporting standard, LOS, population structure), what are the estimated parameters of those factors?

4. The authors have a long time series data of 60 years but it looks like they estimated the average of arithmetic mean of percentage changes of each year. This approach can be useful for understanding short-term fluctuations, but it does not accurately reflect long-term growth trends. This long-term trend should be estimated by taking the logarithm of the outcome (inpatient beds) and regressing it over the time variables (year).

Reviewer #5: This is an interesting and important research study. The main aim is to analyse how the number of beds changed over a 60-year period (1960-2020). The study will benefit if the authors can identify the factors (or explain the reasons) that are behind the reduction in the number of beds in the NHSE. So far, is too descriptive and does not provide information of the drivers of that bed reduction. Therefore, the authors need to clarify that they only find associations between variables rather than causal effects.

Most comments from previous review have been addressed. Though there are some remaining that need clarification. Also, I incorporate additional comments that would improve the manuscript and will ease understanding from readers outside the UK.

Remaining comments from previous review:

1. Comment 14 (Page 14 – “reduction in Geriatric bed capacity by three quarters, and even more if age is controlled for”. Where is this data?). The authors answered, “This is shown in Appendix 1.”.

• Is Appendix 1 Supplementary file 1? This is a Figure. Maybe they referred to Supplementary file 2 where there is a table showing “The number of five different categories of NHS beds and the provision of beds per 100,000 population in England in 1960 and 2019/20.” However, nothing is mentioned about controlling for age besides commenting that Geriatric figures only include 65+ patients and the other categories total population.

2. Comment 16a (Over the years, the understanding and approach to treating and supporting individuals with learning disabilities have evolved significantly. How could these changes have had a direct impact on the provision and reduction of inpatient beds for this population? Would this explain all the bed reduction?). The authors replied, “The paragraph at the end of page 9 addresses this point and we have added the following: “The understanding and approaches to supporting people with Learning Disability has evolved significantly over the past 60 years, and could explain a significant proportion of the bed reductions seen here over the time period.””.

• I think the authors should make these comments if they are supported by the literature. Otherwise, it looks authors’ speculation with no evidence backup. I suggest that the authors include evidence that supports this. This can be explained by changes in the health policy by offering more services at the community level such as in the NHS Five Year Forward Review or the NHS Long Term Plan.

3. Comment 17 (The authors state that there are two main limitations of the study: NHS England data only and lack of information on number of inpatient hospital beds. However, the authors only publish average number of hospital beds per year per category, which raises the question of what additional data could have been controlled for to enhance the study’s robustness. Did the authors not have access to additional data? If not, please state. Otherwise, important variables should be address, such as: socioeconomic factors, population density, urbanization, chronic disease prevalence, and healthcare access (e.g., number of healthcare providers per capita and availability of outpatient services) (…)). The authors responded “Data on Gross Domestic Product and annual change in this has been added to the analysis as well as the the number of deaths and births. We have also added data on Day Only beds, length of stay, the number of inpatient episodes of care, occupied bed days from 1998 onwards to help address the points raised. Reference has been added along with “including psychiatric beds” added to the introduction. The reference section has also been reviewed as suggested with five addition references added and some removed.”.

• I don´t think the authors incorporated the comments suggested in previous reviews. So far, the study is very descriptive with little information to understand the main drivers of this reduction in the number of beds. It could be understood that there was a disparity in the population growth and the number of available beds, but this is unclear. If the authors are using HES data, they can provide more information controlling for socioeconomic factors, for example. Also for Charlson co-morbidities. The authors should think about this question and answer the question addressed by reviewers.

Additional comments:

4. The document provided includes a total of 31 pages. But from page 15, page numbers do not continue previous numbering. Can the authors revise this issue, please?

5. There is a typo on page 3. It says COID19 and should say COVID-19.

6. On page 3 the authors should mention why the ratio of deaths to inpatient bed is important for international comparisons.

7. Page 4 – Material and Methods – The authors mention that they included Day Only beds. Could you explain what kinds of procedures include this concept? I think they are linked to outpatient care (OPC) procedures, but I think this could be clarified, for example if you need to be admitted the day before or you can undergo the procedure in the same day and you will be discharge in few hours during the day.

8. Page 5 – The authors mentioned on page 4 that they got the numbers of inpatient beds following a data request. However, they mention on page 5 that their main data source is HES. Can you clarify this? Also, which HES module you asked for APC, OPC, ACC,…? You can briefly describe each module in 1-2 sentences.

9. Page 5 – how many hospitals are you including? In England there are more than 200 NHS hospitals and some private providers. Since you are also discussing the participation of the private (independent) sector, it would be good to know the proportion in hospitals in England.

10. Page 6 – In the Statistical analysis section, the authors mention “The annual percentage change (APC) in bed provision was the variable used to measure rate of change”. Is this relative to population change or relative change to previous year?

11. Page 7 – RCS & LOESS methods –These methods only provide information on potential associations between variables, not causal effects. Therefore, the authors need this to be clear. Moreover, it would be good to include a Supplementary material where the authors explain briefly how the RCS operates (i.e. using Lusa L, Ahlin Č. Restricted cubic splines for modelling periodic data. PloS One. 2020 Oct 28;15(10):e0241364. Doi: 10.1371/journal.pone.0241364). For the LOESS methods, I suggest the authors clarify this is a function that smooths fitted values.

12. Page 8 – there is a typo in the 3rd paragraph. Where it say “The largest reductions in numbers…”, then it should say “The largest reduction in the number of beds”

13. Page 8 – I suggest the authors move the paragraph above Day Only beds on page 9 to page 8 above the paragraph starting “At the start of the study Acute beds and Mental Illness beds (…)”.

14. Page 9 – regarding Geriatric beds, what is the proportion of 65+ patients in the population and the sample?

15. Page 11 – Figure 3 shows declining and accelerating periods. The accelerating bed closures in 1980-1990 was briefly explained in the Discussion (page 7 +15). What about the second period 2000-2010? I am sure there were changes in the provision of the service at community level but also health budget cuts that may contribute to this decision. Could you compare bed closures to population growth in those years?

16. Page 12 – last paragraph. Please include a gap when you say, “11 million”. Are you saying that the 11% reduction in Occupied beds is explained by halving the average lengths of admissions? I think you wanted to say length of stay of admitted patients. Is this explained by new procedures/treatments or less severe diagnoses?

17. Page 14 – what is the proportion of beds provided from the Independent (private) sector compared to the NHS provision?

18. Page 1 (+15) – From Table 2 onwards, the authors should renumber continuing from page 15.

19. Page 2 (+15) – Table 4 includes p-values. It is suggested that the table incorporates standard errors. The Table incorporates p-values and highlights coefficients that are statistically significant from the Poisson regression. Which variables were included in the regression model? Moreover, the authors may consider replacing p-values with stars, indicating the significance level in the coefficients (*0.05 significance level, **0.01 and ***0.001). That will simplify the understanding of the Table. Did the authors control for providers fixed effects to account for heterogeneity?

20. Page 3 (+15) – There is a typo in the first paragraph. Where it says, “This ratio was much higher in London (440, N=1) then…”. This “then” should be replaced with “than”.

21. Page 7 (+15) – As I mentioned in Comment 15, please discuss the second period of accelerated bed closures 2000-2010.

22. Page 11 (+15) – Strengths and limitations. I am not sure if a 60-year period is a strength. In such long period there is too much heterogeneity caused by different economic cycles and changes in regulation. I would say that using HES data is a strength if the authors justify this rich data with analyses incorporating demographics and other relevant socioeconomic controls. But the authors should make clear if they are using HES in their analysis. So far, it is not clear.

7. PLOS authors have the option to publish the peer review history of their article (what does this mean? ). If published, this will include your full peer review and any attached files.

**Do you want your identity to be public for this peer review?** For information about this choice, including consent withdrawal, please see our Privacy Policy .

Reviewer #4: No

Reviewer #5: No

---

## [Editor Report · Decision Letter 2]

8 Aug 2025

Long term trends in NHS inpatient bed provision in England, 1960-2020

PONE-D-24-12261R2

Dear Dr. Weich,

We’re pleased to inform you that your manuscript has been judged scientifically suitable for publication and will be formally accepted for publication once it meets all outstanding technical requirements.

Kind regards,

Ben Green

Academic Editor

PLOS ONE

Additional Editor Comments (optional):

I have recommended acceptance of this paper. Thank you for your revision.
---

## [Editor Report · Acceptance letter]

PONE-D-24-12261R2

PLOS ONE

Dear Dr. Weich,

I'm pleased to inform you that your manuscript has been deemed suitable for publication in PLOS ONE. Congratulations! Your manuscript is now being handed over to our production team.

Kind regards,

on behalf of

Professor Ben Green

Academic Editor

PLOS ONE